# Hydroxamic acid-modified peptide microarrays for profiling isozyme-selective interactions and inhibition of histone deacetylases

Carlos Moreno-Yruela [1], Michael Bæk [1], Adela-Eugenie Vrsanova[1,2,4], Clemens Schulte [3], Hans M. Maric [3✉] & Christian A. Olsen [1✉]

Histones control gene expression by regulating chromatin structure and function. The posttranslational modifications (PTMs) on the side chains of histones form the epigenetic landscape, which is tightly controlled by epigenetic modulator enzymes and further recognized by so-called reader domains. Histone microarrays have been widely applied to investigate histone–reader interactions, but not the transient interactions of $Zn^{2+}$-dependent histone deacetylase (HDAC) eraser enzymes. Here, we synthesize hydroxamic acid-modified histone peptides and use them in femtomolar microarrays for the direct capture and detection of the four class I HDAC isozymes. Follow-up functional assays in solution provide insights into their suitability to discover HDAC substrates and inhibitors with nanomolar potency and activity in cellular assays. We conclude that similar hydroxamic acid-modified histone peptide microarrays and libraries could find broad application to identify class I HDAC isozyme-specific substrates and facilitate the development of isozyme-selective HDAC inhibitors and probes.

[1] Center for Biopharmaceuticals & Department of Drug Design and Pharmacology, University of Copenhagen, Universitetsparken 2, DK-2100 Copenhagen, Denmark. [2] Institute of Applied Biosciences & Department of Food Chemistry and Toxicology, Karlsruhe Institute of Technology, Adenauerring 20a, D-76131 Karlsruhe, Germany. [3] Rudolf Virchow Center, Center for Integrative and Translational Bioimaging, University of Würzburg, Josef-Schneider-Str. 2, D-97080 Würzburg, Germany. [4] Present address: Division of Proteomics of Stem Cells and Cancer, DKFZ German Cancer Research Center, Im Neuenhemier Feld 581, D-69120 Heidelberg, Germany. ✉email: hans.maric@virchow.uni-wuerzburg.de; cao@sund.ku.dk

The human genetic code is stored in nucleosome particles, supramolecular assemblies of DNA wrapped around histone octamers, which comprise two copies of each of the core histone proteins H2A, H2B, H3, and H4[1]. Epigenetic marks in the form of posttranslational modifications (PTMs) to histone amino acid side chains critically affect chromatin structure and ultimately gene expression[2,3]. Aberrant gene expression is directly associated with cancer, autoimmune deficiencies, and neurodegenerative diseases and may be modulated by altering the epigenetic state of the cell[2–5]. This epigenetic state is highly dynamic and responds to a variety of external stimuli of the cell such as nutrient composition and stress, but also at a systemic level to diet, lifestyle, environment, and, importantly, pharmaceutical intervention[2,4]. Therefore, a detailed understanding of the enzymes that shape the histone PTM landscape is of great interest. Peptide microarrays have been widely employed for the characterization of histone reader domains as well as histone-modifying enzymes and antibodies[6–13]. Also, chemical protein and peptide synthesis have facilitated the incorporation of modified amino acids to decipher how specific histone PTMs (e.g., acetylation, methylation, phosphorylation, ubiquitylation, and citrullination) affect recognition and chromatin structure at the molecular level[14]. One component of this complex, dynamic regulatory machinery is the reversible acetylation of the ε-amino group of lysine residues, which is regulated by histone acetyltransferases (HATs) and histone deacetylases (HDACs 1–11; Fig. 1a)[15,16]. Further, the resulting ε-N-acetyllysine (Kac) mark is specifically recognized through bromodomains, constituting an important class of reader domains[5]. Peptide microarray strategies have proven valuable for providing binding affinities of large numbers of peptides[17,18], and have especially proven efficient for interrogating the specificity of epigenetic reader domains[6,7,9,11,12]. Compared to HATs and bromodomains[8–10,13], however, the application of peptide microarrays to the investigation of HDACs has remained more challenging[19–22]. The fast turnover of acetylated

substrates and the resulting transient HDAC–substrate interaction necessitated coupling of the conventional peptide microarray approach with a secondary readout, such as radioactive isotopes[23], fluorescence tags[24,25], immunodetection of the resulting acetylation states[21,22,26,27], or mass spectrometry[19,20,28,29]. Despite these efforts, the understanding of the inhibitor and substrate selectivity of the human $Zn^{2+}$-dependent HDAC isoforms is still incomplete. This is particularly true for class I HDACs (HDACs 1–3 and 8, Fig. 1a), which are the HDAC isozymes primarily targeted by clinically-used drugs[30,31] and responsible for histone deacetylation[16].

Here, we report the synthesis and application of peptide microarrays that circumvent the aforementioned limitations by the introduction of hydroxamic acid-containing residues in place of the Kac residues (Fig. 1b)[32–34], which thereby provide a platform to investigate HDAC binding preferences in high-throughput (Fig. 1c). Complementary functional assays evaluate the ability of this approach to predict HDAC binding, inhibition, and activity.

## Results

**Modified peptides enable interrogation of HDAC binding in microarray format.** Conventional display of acetylated peptides in microarray format does not allow studying HDAC binding due to the fast turnover and resulting transient HDAC–peptide interaction. We envisioned that modifying acetylated peptides similar to common competitive inhibitors of class I HDACs could enable the capture and detection of HDACs in microarray format. The cap group of such inhibitors interacts with the enzyme surface close to the catalytic pocket and the linker projects a zinc-binding group through a channel to the $Zn^{2+}$ ion in the active site (Fig. 2a)[31,32]. Consequently, the turnover of Kac-containing peptides by $Zn^{2+}$-dependent HDACs can be inhibited by entities where the Kac has been substituted with zinc-binding groups, and peptides containing the hydroxamic acid[32–34] or the o-aminoanilide[35] functionalities have been shown to bind and/or capture HDAC enzymes. Here, we explored the use of

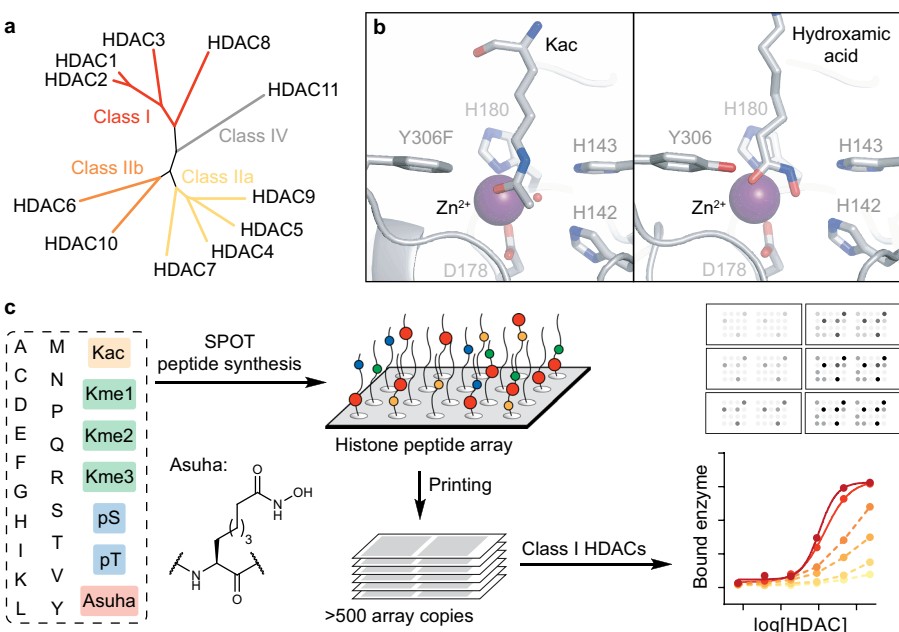

**Fig. 1 Hydroxamic acid-modified microarrays for the direct capture and detection of class I HDAC isozymes. a** Phylogenetic relationship of the 11 human $Zn^{2+}$-dependent HDACs. **b** X-ray co-crystal structures showing the binding pocket of HDAC8 interacting with a Kac-containing peptide substrate or a hydroxamic acid. In contrast to Kac (left, PDB: 5DIC, Y306F mutation required for deactivation), hydroxamic acids (right, PDB: 3EW8) stall the transient interaction by chelating the $Zn^{2+}$ ion in the active site. **c** Workflow. Peptide microarrays are prepared using the appropriate hydroxamic acid-containing aminosuberic acid analog (Asuha). Resulting arrays allow for the direct assessment of HDAC isoform-specific binding. Kme1, Kme2, Kme3: ε-N-mono-, di-, and trimethyllysine; pS: O-phosphoserine; pT: O-phosphothreonine; Kac: ε-N-acetyllysine.

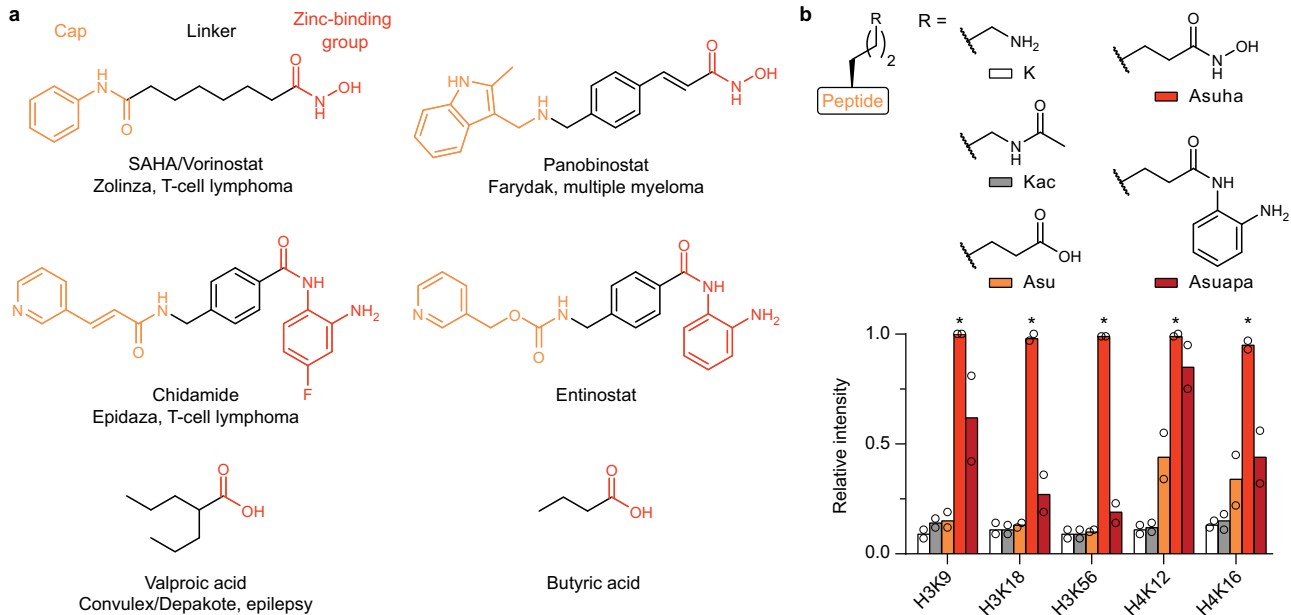

**Fig. 2 Zinc-binding groups allow probing HDACs in microarray format. a** Approved drugs and examples of candidates in clinical trials that target class I HDACs[31]. Shared features include a cap group, a hydrophobic linker region, and zinc-binding group. **b** Display and probing of modified 15-mer histone peptides for capture of HDAC3 (10 nM) on microarray slides. Incorporation of a 2-aminosuberic acid hydroxamic acid derivative (Asuha) or a 2-aminosuberic acid o-aminoanilide derivative (Asuapa) but not the carboxylic acid (Asu) allow probing of HDAC binding in microarray format ($n = 2$ independent experiments). Source data are provided as a Source Data file. *Saturated chemiluminescence signal.

these moieties in histone tail peptide microarrays to interrogate the binding of class I HDACs. Modified peptide microarrays were produced using μSPOT, a variant of the widely applied SPOT method[36–38]. We found that replacing lysine residues in histone sequences by L-2-amino-8-(hydroxyamino)-8-oxooctanoic acid (Asuha) or L-2-amino-8-((2-aminophenyl)amino)-8-oxooctanoic acid (Asuapa) but not L-2-aminosuberic acid (Asu) or acetylated lysine (Kac) efficiently captured the HDAC–substrate interaction (Fig. 2b). Both Asuha and Asuapa provided sequence-dependent binding results, and Asuha was selected for further investigations due to the higher HDAC retention shown by the corresponding peptides. The length of the histone-based peptides was chosen to be 15 residues, because the synthetic efficiency of the μSPOT method decreases substantially above the 15–16-mer length. Our findings were in line with the observation that class I-selective o-aminoanilides, such as the approved drug chidamide and the clinical trial candidate entinostat, are among the most potent inhibitors, only surpassed by less selective hydroxamic acid-containing compounds such as panobinostat[31].

We next envisioned that hydroxamic acid-modified peptide microarrays would enable the direct and simple interrogation of HDAC isozyme binding and, potentially, facilitate the development of isozyme-selective inhibitors and probes (Fig. 1c). The implementation of this technique required an improved synthetic route for the Asuha building block. Modification of an acetamidomalonate alkylation/enzymatic resolution strategy[39,40] provided an affordable and scalable synthetic route for Fmoc-Asuha(ᵗBu)-OH (2 g, 20% overall yield, Supplementary Fig. 1) compared to previously reported syntheses[33,34,41]. Hydroxamic acid-modified μSPOT arrays were then produced and successful synthesis of the modified peptides was validated by LC-MS analysis of a subset of crude peptides prior to printing of the microarrays (Supplementary Table 1). Unfortunately, the crude peptides could not be triturated before chromatography due to the minute amounts produced and the resulting HPLC traces, therefore, included protecting group byproducts. Nevertheless, an average purity of 41%, corresponding to average coupling efficiency of at least 95%, was observed, which is in line with

standard SPOT[42,43] and μSPOT[44] peptide synthesis. The detection of a dilution series of His-tagged HDAC2 spotted on a μSPOT slide indicated a linear correlation between signal and spotted amount (Supplementary Fig. 2, $r^2 = 0.96$) when a horseradish peroxidase (HRP)-conjugated anti-His antibody was used. Thus, taken together, the μSPOT method allowed for economic production of hundreds of microarray copies from a single round of synthesis and thereby facilitated the semi-quantitative assessment of HDAC isozyme–peptide interactions (Fig. 1c)[37,38].

**Microarray-based screening and fine mapping of class I HDAC interactions.** Next, we explored the use of hydroxamic acid-modified microarrays to study class I HDAC substrate recognition. We displayed a library of 291 histone-based peptides (16 amino acids in length and with a 3-residue frameshift) with the Asuha building block at the position of one lysine residue at a time, in order to comprehensively and systematically study recognition of the four core histones by HDACs 1–3 and 8 (Supplementary Table 2). Microarray copies were titrated with recombinant human His-tagged class I HDACs and binding was detected with an anti-His antibody as before.

The recapitulation of known binding sites demonstrated the robustness of our platform. As expected, only peptides including Asuha residues were able to enrich HDACs. At the same time, we observed striking differences in isozyme affinity and selectivity (Fig. 3). HDACs 1, 2, and 8 displayed high sequence specificity, in agreement with previous substrate selectivity studies on HDAC8[28,45], and also similar to HDAC6[22]. In contrast, HDAC3 bound more promiscuously and to sequences located both at the core and the tails of the histones. This might indicate that HDAC3 is less sensitive to the peptide sequence, as also reported for short peptide substrates[46,47] and acetylated peptide microarrays[20]. Sequence logo plots were generated to highlight the amino acid preferences of each HDAC (Supplementary Fig. 3). These plots revealed an overall preference for Arg and Lys

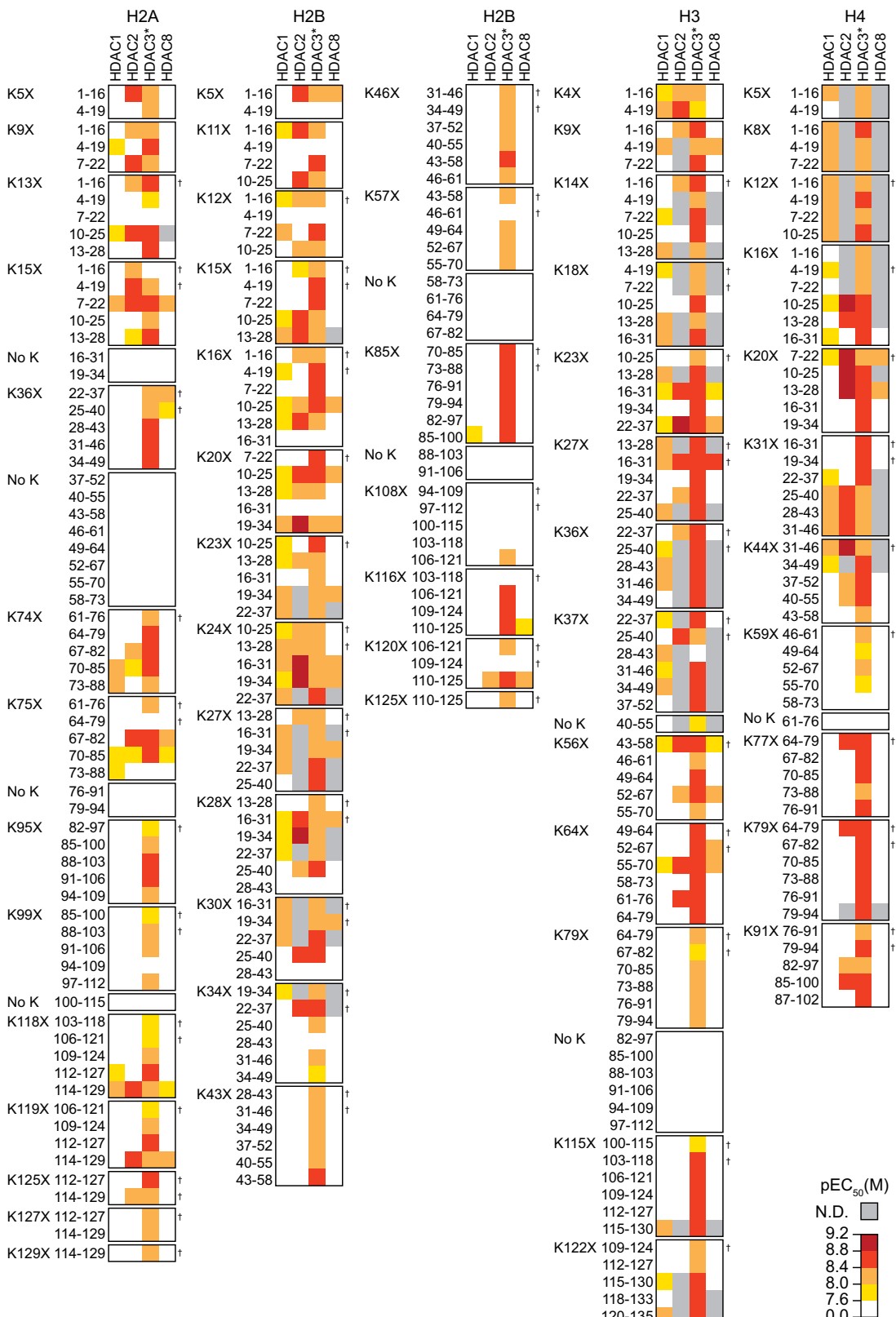

**Fig. 3 Screening of class I HDAC–histone peptide binding in microarray format.** Heat map representation of relative HDAC binding affinity for Asuha (X)-containing histone peptides. The pEC$_{50}$ values represent the concentration of HDAC enzyme required for half-maximal binding signal ($n = 4$ independent dose-response experiments). White squares represent weak binders (pEC$_{50}$ < 7.6). N.D. = not determined (non-sigmoidal data). Source data are provided as a Source Data file. *HDAC3 tested in combination with the deacetylase activating domain (DAD) of NCoR2. †Matrix steric effects might impede protein binding to these peptides.

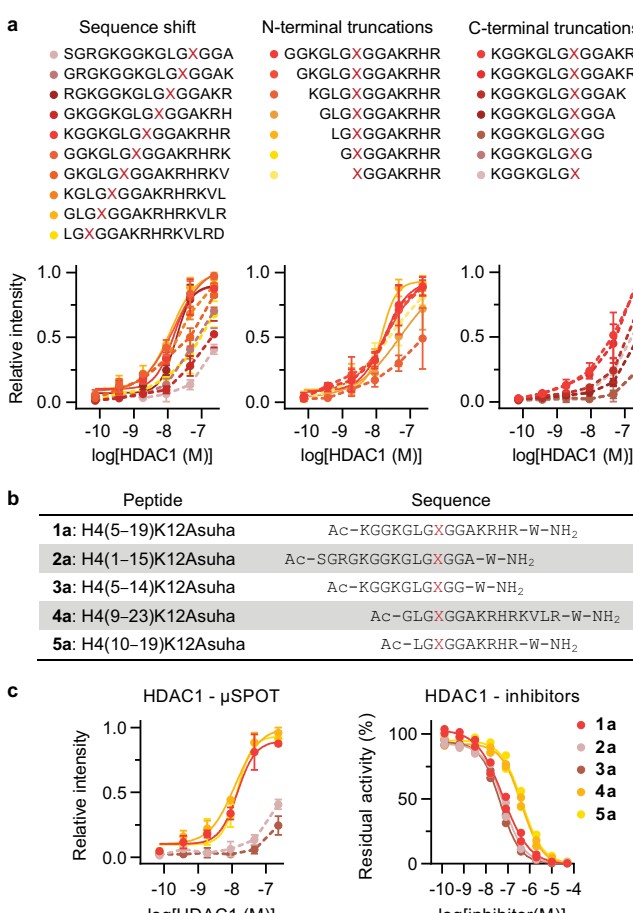

**Fig. 4 Application of hydroxamic acid-modified peptide microarrays for mapping peptide binding requirements. a** Fine mapping of relative HDAC1 binding affinities in microarray format (data represent mean ± SEM, n = 4 independent experiments). See Supplementary Figs. 4–7 for complete data sets for HDACs 1–3 and 8. X: Asuha. Source data are provided as a Source Data file. **b** Resynthesized peptide sequences including the parent peptide (**1a**), two frame-shifted sequences (**2a**, **4a**), a C-truncated sequence (**3a**), and a N-truncated sequence (**5a**). **c** Comparison of the inhibitory curves of resynthesized peptides **1a**–**5a** with their corresponding microarray binding curves against HDAC1 (microarray data represent mean ± SEM, n = 4 independent experiments; inhibition data represent n = 2 independent experiments). See Supplementary Fig. 8 for data on HDACs 2, 3 and 8 and for half-maximal inhibition concentrations. Source data are provided as a Source Data file.

residues at the positions flanking the Asuha residue for all class I HDACs, which was previously observed for HDAC8[28] and, to some extent, also HDAC3[20]. Interestingly, HDAC6 shows the opposite preference[22]. Further class I HDAC selectivity was observed for sequences containing Gly at the -2 and -3 positions, and Arg and Lys at the −4, +3, and +4 positions. Not surprisingly, histone tails are rich in Arg, Lys, and Gly residues, which could explain the obtained sequence preference. In this regard, we expect that future evaluation of potential isozyme-specific preferences would benefit from the use of randomized peptide arrays.

The microarrays were also applied for mapping binding sites with semi-quantitative estimation of binding affinities. This was outlined for histone H4 lysine 12 (H4K12) (Fig. 4a). Titration of single amino acid frame-shifted peptides and single amino acid truncation variants suggested exact binding sites and indicated that the C-terminus of the selected peptide is required for binding

to HDACs 1, 2, and 8 (Supplementary Figs. 4–7). Peptides **1a**–**5a** were then synthesized, purified, and tested as HDAC inhibitors to validate these observations. Peptides **4a** and **5a** showed 10-fold lower potency than the parent peptide (**1a**), as suggested by the data on HDACs 2 and 8, but not HDAC1 (Fig. 4c and Supplementary Fig. 8). More strikingly, peptides **2a** and **3a** with C-terminal truncations retained potency against all HDACs. This could indicate that particularly HDACs 1, 2, and 8 are sensitive to the steric effect of the microarray matrix and struggle to bind to Asuha residues closer than 5 amino acids to the C-terminal peptide conjugation site. On the other hand, HDAC3 was not affected by this effect, which could also explain the higher rate of positive results for this isozyme. Overall, the mapping of the H4K12 site revealed a complex sequence requirement for HDACs 1, 2, and 8, especially for residues within ±7 positions relative to the acetylated lysine. This mapping highlighted the importance of steric effects introduced by the immobilization and thus the importance to validate results with resynthesized peptides in solution.

**Microarray-based study of the binding effects of posttranslational modifications.** The action of epigenetic modulators, including HDACs, is orchestrated by PTMs[48]. We next applied our microarray approach to resolve how specific PTM combinations control isozyme-specific class I HDAC function. To probe this, we selected two H3 N-terminal peptides for modification in a combinatorial manner according to reported histone PTMs (Supplementary Table 4)[48,49]. The included modifications were O-phosphoserine (pS), O-phosphothreonine (pT), ε-N-mono-, di-, and trimethyllysine (Kme1, Kme2, and Kme3, respectively), Kac, and ε-N-thioacetyllysine (Kthioac), which is a Kac analog with enhanced stability towards hydrolases[50,51] that is only cleaved to some extent by HDACs 6 and 8[52] and class I sirtuins[53]. Remarkably, the microarray approach resolved complex and highly isozyme-specific binding profiles (Fig. 5a). Importantly, binding of HDACs to H3 peptides was diminished by phosphorylation of residues Ser10 or Thr11. Conversely, enrichment of HDACs 1, 2, and 8 was improved by modification of the Lys4 and Thr6 residues. Modifications at Lys23 also diminished HDAC8 binding, although its proximity to the C-terminus makes it difficult to interpret this trend due to the aforementioned matrix steric effects. Notably, the pronounced effect of phosphorylation of H3S10 in the microarray may mirror an altered recognition by epigenetic enzymes shown to play a pivotal role in the histone modification cross-talk[54,55]. To further validate the prediction of HDAC isozyme substrate recognition and selectivity, we studied deacetylation of resynthesized peptides modified at the Lys4 and Ser10 residues. Thus, H3 sequences were prepared as Kac-modified peptides with and without thioacetylation and phosphorylation by standard Fmoc/tBu SPPS (Fig. 5b) and were incubated with recombinant class I HDACs. Substrate conversion was followed by LC–MS analysis, which revealed that Ser10 phosphorylation, as predicted by the microarray, decreased the deacetylation by HDACs, whereas Lys4 thioacetylation did not promote conversion by HDACs 1, 2, and 8 (Fig. 5b and Supplementary Fig. 9). Taken together, our results highlight the applicability of hydroxamic acid-modified microarrays to interrogate changes in HDAC substrate that give rise to substantially altered recognition in high-throughput. In particular, we demonstrated that these microarrays may help identify individual positions and specific PTMs that critically alter peptide binding affinity and isozyme selectivity; thus, providing a useful tool to study HDAC function and its regulation.

**Identification of potent HDAC inhibitors with tailored isozyme selectivity.** Multiple natural and synthetic inhibitors of class

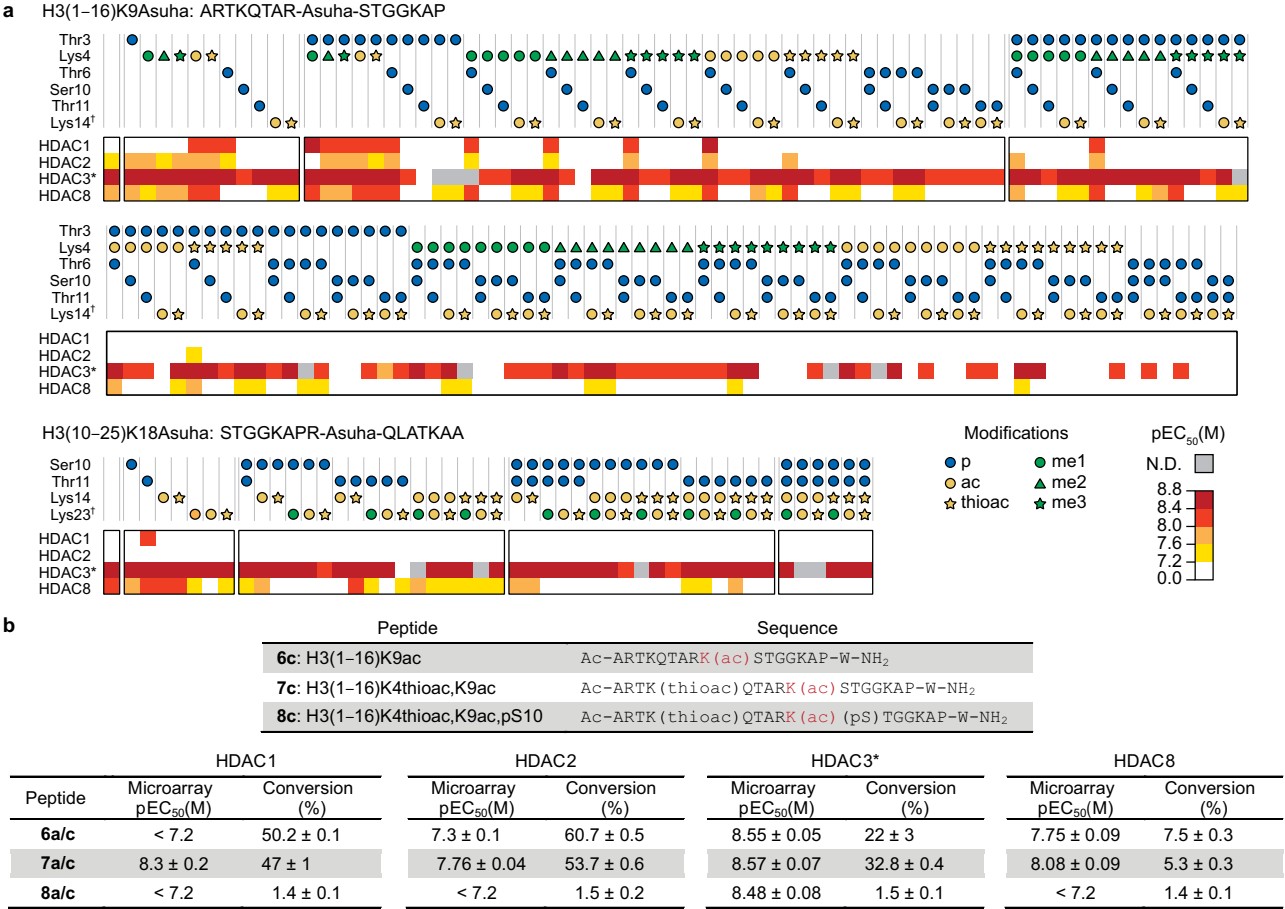

**Fig. 5 Deciphering the regulation of HDAC action by histone PTMs. a** Heat map depicting effects of neighboring chemical modifications on the affinity of peptides H3(1–16)K9Asuha and H3(10–25)K18Asuha. p: phosphorylation, ac: acetylation, thioac: thioacetylation, me1: monomethylation, me2: demethylation, me3: trimethylation. Combinations of 4, 5, and 6 modifications abolished binding to H3(1–16)K9Asuha peptides and have not been included. The $pEC_{50}$ values represent the concentration of HDAC enzyme required for half-maximal binding signal, $n = 4$ independent experiments). White squares represent weak binders ($pEC_{50} < 7.2$). Source data are provided as a Source Data file. **b** Resynthesized peptide sequences with Kac at the position of study, and relative deacetylation by class I HDACs. Enzyme and substrate were incubated for 1 h at 37 °C, and relative conversion was determined by LCMS (see Supplementary Fig. 9 for sample assay traces). Microarray data represent mean ± SEM, $n = 4$ independent experiments; conversion data represent mean ± SD, $n \geq 2$ independent experiments. *HDAC3 tested in combination with the DAD domain of NCoR2. †Matrix steric effects might contribute to the effects at these positions.

I HDACs (HDACs 1–3 and 8) have been reported, some of which are approved for clinical use[30,31]. The majority are small molecules bearing a hydroxamic acid moiety, which exhibit limited isozyme selectivity. Microarray screening (Figs. 3 and 5) identified modified peptides with potential isozyme selectivity and suggested that even small chemical modifications can exert strong and isozyme-selective effects on HDAC recognition. To validate the reported selectivity and to quantify their inhibitory potency towards class I HDACs, we resynthesized additional nine hydroxamic acid-containing peptides (**9a–17a**) from the initial screen and evaluated HDAC inhibition in an in vitro fluorogenic assay[56,57]. All tested peptides displayed robust nanomolar potency against HDACs 1–3, and also against HDAC8 in the case of peptides **6a**, **13a**, and **14a** (Fig. 6a and Supplementary Fig. 10). As expected, control peptides with a lysine residue instead of Asuha did not inhibit any of the tested HDACs, even at high micromolar concentration.

The determined inhibitor profiles confirm the potency of multiple hits and validate the concept that peptide scaffolds can achieve isozyme selectivity, but they also reveal that potent inhibitors may be overlooked during microarray screening. Sequences **11a**, **12a**, and **13a** exhibited selectivity for HDAC3

in the microarray (Fig. 3). For the resynthesized peptides, we found ~4-fold, ~2-fold, and >8-fold selectivity for HDAC3 vs. HDACs 1 and 2, respectively (Fig. 6c, d), which supports the use of our screening technique to discover selective lead inhibitors. Peptide **11a** was also >60-fold selective for HDAC3 vs. HDAC8, whereas peptide **13a** inhibited HDAC8 potently, which was not anticipated. Interestingly, peptides **9a** and **16a** displayed over 100-fold selectivity against HDACs 1–3 versus HDAC8 (Fig. 6c, d), which is uncommon among hydroxamic acid-containing inhibitors[57]. Conversely, peptide **14a** inhibited all four isozymes with similar potency, in agreement with the microarray data (Figs. 3 and 6d). The cytosolic isozyme HDAC6 was also included in the profiling because it is inhibited by several known hydroxamic acid-containing HDAC probes[57]. Peptides **1a** and **15a** were the most and least potent inhibitors against this isozyme, respectively (**1a**: $K_i = 5.31 \pm 0.03$ nM, **15a**: $K_i = 114 \pm 2$ nM, Supplementary Fig. 11a). As expected, the inhibitors were highly selective over class IIa HDACs, represented by HDAC4, which was only partially inhibited at inhibitor concentrations of up to 50 μM (Supplementary Fig. 11b).

Though a number of the potencies recorded for resynthesized peptides recapitulated the microarray findings, the potencies of a

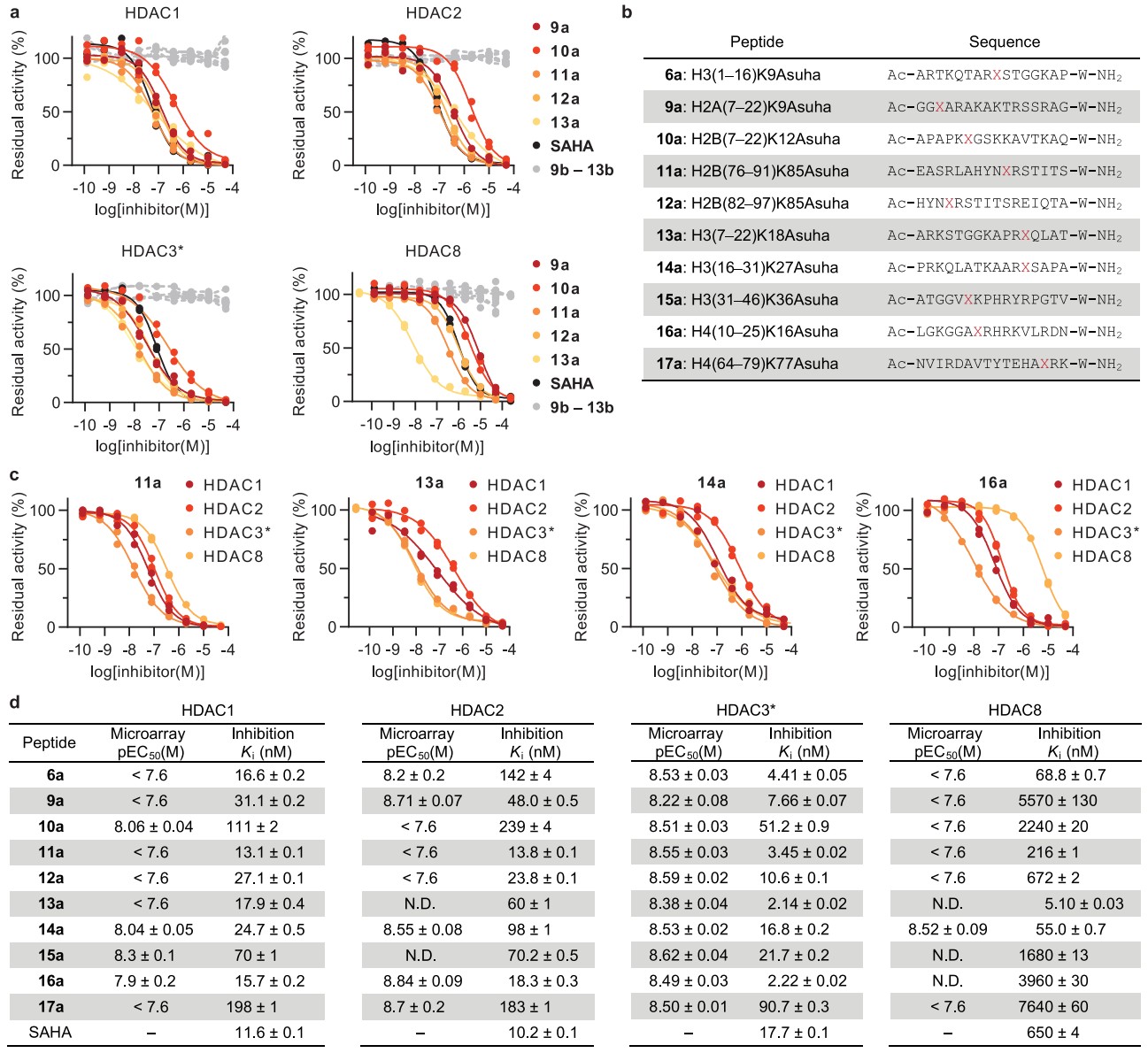

**Fig. 6 Identification of potent HDAC inhibitors with distinct selectivity profiles. a** Dose-response HDAC inhibition curves for Asuha (X)-containing peptides (**9a**–**13a**), the HDAC inhibitor SAHA and lysine-containing controls (**9b**–**13b**). See Supplementary Fig. 10 for curves corresponding to **6a**, **14a**–**17a**, and corresponding controls. **b** Sequence of resynthesized peptides. **c** Dose-response curves for peptides **11a**, **13a**, **14a**, and **16a**. **d** Summary of microarray screening data and the determined potency of resynthesized Asuha-containing peptides in solution. See Supplementary Fig. 12 for visual representation. *HDAC3 tested in combination with the DAD domain of NCoR2. All microarray data represent mean ± SEM, $n = 4$ independent experiments; all inhibition data represent $n = 2$ independent experiments. Source data are provided as a Source Data file.

number of the weaker binders were not estimated accurately in the microarray screening, thereby highlighting the need for detailed follow-up validation when applying this technology (Supplementary Fig. 12). Peptides **13a** and **17a**, which appeared to exhibit preference for HDACs 2 and 3 over HDAC1 in the array experiments, were both equipotent across all three recombinant HDACs 1–3, when tested in solution. This highlighted once more the importance of sufficient C-terminal spacing in microarray format, as both peptides included the Asuha residue within 5 amino acid positions of the linker. In addition, peptide **11a**, which was among the most potent inhibitors of HDACs 1 and 2, did not show tight binding on the array for these two isozymes. Nevertheless, microarray screening enabled the discovery of potent peptide scaffolds with distinct profiles of HDAC inhibition. Additionally, our data confirmed a major contribution of the peptide sequence to

substrate turnover, inhibitor potency, and isozyme selectivity (Fig. 6d and Supplementary Fig. 13), indicating that peptide scaffolds are an appropriate source for isozyme selective inhibitors and affinity probes.

**Hydroxamic acid-modified peptides exhibit activity in cells.** Next, we aimed to explore the cellular activity of selected HDAC inhibitors. We chose peptides **1a** and **13a**, which exhibit nanomolar potency against class I HDACs and HDAC6 and, at the same time, harbor multiple basic residues that could facilitate cell permeation. HEK293T cells were treated with the peptides for 5 h, and subsequent cell lysis and Western blot analysis revealed upregulation of acetylation of both tubulin and histones compared to DMSO control (Fig. 7a). These results strongly indicate that peptides **1a** and **13a** effectively inhibit the cytosolic HDAC6

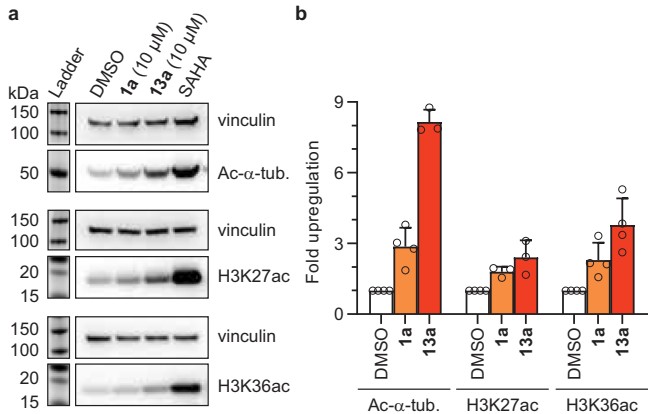

**Fig. 7 Inhibition of HDACs in cultured cells. a** Sample Western blots on tubulin acetylation (Ac-α-tubulin) and histone acetylation (H3K27, H3K36) upon 5 h treatment with peptides **1a** and **13a** at 10 μM concentration, with SAHA as positive control (Ac-α-tubulin: 1 μM; H3K27ac, H3K36ac: 5 μM). See Supplementary Fig. 14 for full blots of all replicates. **b** Quantification of acetylation, normalized to vinculin loading control, and relative to DMSO (data represent mean ± SD, n = 4 biologically independent samples). Source data are provided as a Source Data file.

and the nuclear HDACs 1–3 in the cell, which is in agreement with their in vitro inhibition profiles against recombinant HDACs[15]. In particular, peptide **13a** induced an 8-fold increase in tubulin acetylation and a 4-fold increase in H3K36 acetylation (Fig. 7b). Thus, the high-throughput screening and subsequent evaluation in solution delivered cellularly active HDAC probes.

## Discussion

Peptide microarrays are broadly applied to study writers and readers of histone modifications[6,7,9,11,12], especially bromodomains[8–10,13]. Their application to erasers of histone acetylation, however, has remained limited to activity-based approaches, as microarray binding experiments required secondary readouts[19–29]. In addition, activity-based approaches rely on antibodies specific for a certain peptide modification, which can sometimes lead to biases induced by the sequence context. Here, we describe a facile, cost-effective, and scalable protocol for the preparation of the necessary hydroxamic acid building block and demonstrate its use to produce hydroxamic acid-modified peptide microarrays. These arrays allow the simple and direct profiling of HDAC binding and inhibition with minimal sample consumption and with the use of extensively validated tag-specific antibodies. When used systematically, similar arrays could provide large data sets and answer fundamental questions related to HDAC action and its regulation. Here, we demonstrated the use of this format to identify individual positions and modifications that determine isozyme-specific binding, and to the discovery of cellularly active peptides with tailored HDAC selectivity.

Hydroxamic acid-modified peptides have been used as substrate mimics and were shown to recapitulate important aspects of HDAC substrate recognition[32–34,58]. Here, we systematically explored the sequence and PTM requirements for the recognition of histone tails, as represented by Asuha-modified peptides, by HDACs 1, 2, 3, and 8. Our extensive dataset will enable future in-depth investigation of specific PTM effects on HDAC activity. These follow-up studies would benefit from the use of synthetic histones and reconstituted nucleosomes, which can reveal sequence effects beyond the length limitations of synthetic peptides[45,58].

Selected inhibitor candidates displayed nanomolar potency and distinct selectivity profiles. Yet, the binding hierarchies in the screens did not represent the exact inhibitor selectivity profiles

defined by more accurate in-solution assays. The potency of peptides with the Asuha residue close to the C-terminus was not well predicted, likely due to steric interference of the microarray matrix, and a number of additional potent inhibitors were overlooked by microarray screening. Kinetic insight from peptide macrocycles[59] and an X-ray co-crystal structure of HDAC1 with a hydroxamate-containing H4-based peptide[32] suggest that Asuha-containing peptides bind in the active site. Nonetheless, future studies should explore to what extent HDAC substrate turnover correlates with inhibitor potency or selectivity. This question could be addressed using the approach presented herein, together with alternative modifications such as the here described Asuapa building block. Such studies should take into account isozyme-specific effects, kinetic parameters, and contributions from additional binding partners[60]. Moreover, incubation with recombinant HDACs bearing active site mutations or co-incubation with validated HDAC probes could improve the selection of competitive inhibitor candidates from the array screening.

Selected microarray hits proved to penetrate the cell membrane and elicit hyperacetylation of class I and class IIb HDAC targets. Thus, combined with structural optimization such as stapling and/or N-methylation[61,62], promising peptide-based HDAC probes could be afforded. The approach presented here could also be applied to the study class IIa HDACs, taking advantage of trifluoromethyl ketone-containing building blocks[63], and deacetylase enzymes from other species[64–67]. The HDAC probes here reported may be the first of a new class of versatile tools for investigating class I HDACs and, importantly, also their native protein complexes[33].

Further, the employed principle may be applied to alternative peptide microarray types including high-density lithography[68] and laser-based on-chip synthesis approaches[69,70], thereby further increasing the throughput. In addition, alternative binding detection methods could improve the dynamic range of the screening assay and potentially enhance the predictive value of the array[18]. Probing such microarrays with cellular lysates, using labeled HDACs and complex partners or highly specific antibodies, may provide a direct readout for the substrate-binding requirements of the native complexes[16]. Combination of such readout with different cells and tissues, as well as with perturbed HDAC levels, could greatly improve our current understanding of HDAC action in vivo.

Finally, we envision that the current approach can facilitate the development of selective chemoproteomic probes that would complement the already applied unspecific HDAC-targeting probes[71–73]. We expect that this work will enable the discovery, characterization, and selective targeting of cellularly active and, potentially, therapeutically relevant multiprotein complexes formed by the different HDAC isoforms[16].

## Methods

**Chemical Synthesis.** General synthetic methods, experimental procedures, and compound characterization can be found in the Supplementary Methods.

**Assay Materials.** Assays were performed in 4-(2-hydroxyethyl)-1-piperazineethanesulfonic acid (HEPES) buffer [50 mM HEPES/Na, 100 mM KCl, 0.001% (v/v) tween-20, 0.2 mM tris(2-carboxyethyl)phosphine (TCEP), 0.5 mg/mL bovine serum albumin (BSA), pH 7.4] unless otherwise stated. Recombinant HDAC enzymes employed in μSPOT, inhibitor, and substrate assays were from commercial sources: HDAC1 (full length, C-terminal His tag, C-terminal FLAG tag, BPS Bioscience, cat. #: 50051, lots 170105-1 and 181108-1, purity ≥ 79%, activity ≥ 460 pmol/min/μg), HDAC2 (full length, C-terminal His tag, BPS Bioscience, cat. #: 50002, lots 160701 and 160630, purity ≥ 84%, activity ≥ 675 pmol/min/μg), HDAC3/NCoR2 (full length, C-terminal His tag, NCoR2 N-terminal GST tag, BPS Bioscience, cat. #: 50003, lots 130819 and 190327, purity ≥ 80%, activity ≥ 3000 pmol/min/μg), HDAC8 (full length, C-terminal His tag, BPS Bioscience, cat. #: 50008, lots 150714 and 161216, purity ≥ 90%, activity ≥ 300

pmol/min/µg), HDAC4 (aa 627–1084, BPS Bioscience, cat. #: 50004, lot 130828-G, purity ≥ 89%, activity ≥ 103225 pmol/min/µg), HDAC6 (full length, BPS Bioscience, cat. #: 50056, lot 151130-C, purity ≥ 88%, activity ≥ 150 pmol/min/µg). Antibodies: 6x-His tag monoclonal antibody (HIS.H8), HRP-conjugated (ThermoFisher, cat. #: MA1-21315-HRP).

**µSPOT Microarray Preparation.** Peptides were synthesized using a ResPep SL synthesis robot equipped with a CelluSPOT synthesis module (Intavis AG, Germany). Parallel synthesis was carried out on acid-soluble Fmoc-β-Ala-etherified cellulose disks (area 0.12 cm$^2$, loading 1.0 µmol/cm$^2$, Intavis AG), by Fmoc/$^t$Bu peptide synthesis on cellulose support (SPOT synthesis). The following protected amino acids were employed: Fmoc-Ala-OH, Fmoc-Arg(Pbf)-OH, Fmoc-Asu($^t$Bu)-OH[39], Fmoc-Asuapa (Boc)-OH (**S8**), Fmoc-Asuha($^t$Bu)-OH (**S5**), Fmoc-Asn(Trt)-OH, Fmoc-Asp(O$^t$Bu)-OH, Fmoc-Cys(Trt)-OH, Fmoc-Gln(Trt)-OH, Fmoc-Glu($^t$Bu)-OH, Fmoc-Gly-OH, Fmoc-His(Trt)-OH, Fmoc-Ile-OH, Fmoc-Leu-OH, Fmoc-Lys(Ac)-OH, Fmoc-Lys(Me, Boc)-OH, Fmoc-Lys(Me)$_2$-OH, Fmoc-Lys(Me)$_3$-OH, Fmoc-Lys(thioac)-OH, Fmoc-Lys (Boc)-OH, Fmoc-Pro-OH, Fmoc-Ser($^t$Bu)-OH, Fmoc-Ser(PO$_3$BzlH)-OH, Fmoc-Thr ($^t$Bu)-OH, Fmoc-Thr(PO$_3$BzlH)-OH, Fmoc-Trp(Boc)-OH, Fmoc-Tyr($^t$Bu)-OH and Fmoc-Val-OH. Peptide elongation was achieved by double coupling in DMF (20 and 15 min) with Fmoc-Xaa-OH/Oxyma/DIC (1.0:1.1:1.5 equiv) in at least 2-fold excess, followed by capping with acetic anhydride. Functionalized cellulose disks were transferred to 1.2 mL 96-tube racks (Intavis AG, cat. # 32.199) and shaken with TFA–CH$_2$Cl$_2$–H$_2$O–$^i$Pr$_3$SiH, [150 µL, 80:12:5:3 (v/v)] for 2 h at room temperature. Solutions were then removed, and disks were suspended in TFA–CF$_3$SO$_3$H–H$_2$O– $^i$Pr$_3$SiH [250 µL, 88.5:4:2.5:5 (v/v)] and shaken overnight at room temperature. Thereafter, solutions were triturated with ice-cold diethyl ether (700 µL), stored at −20 °C for 2 h and centrifuged at 1000 rcf (45 min, 4 °C). Supernatants were discarded, and pellets were rinsed with diethyl ether in the same manner. Then, 96-tube racks were ventilated for 2.5 h to allow for residual diethyl ether to evaporate, and pellets were dissolved in DMSO (500 µL) and stored at −20 °C. µSPOT microarrays were prepared by printing DMSO Stock/SSC buffer mixtures (30 nL, 2:1, saline sodium citrate buffer: 15 mM sodium citrate, 0.1 M NaCl, pH 7) in duplicate on polystyrene-coated microscope glass slides (Intavis AG, cat. # 54.112) with a Slide Spotter Robot (Intavis AG).

**Analysis of Cleavable µSPOT Peptides.** Disks containing Rink amide linkers were incubated in 1.5 mL plastic tubes with TFA–2,2′-(ethylenedioxy)diethanethiol (DODT)–H$_2$O–$^i$Pr$_3$SiH [250 µL, 92.5:2.5:2.5:2.5 (v/v)] for 2 h at room temperature. Solutions were transferred to new plastic tubes, triturated with ice-cold diethyl ether (750 µL), stored at −20 °C for 1 h and centrifuged at 17968 g (30 min, 4 °C). Thereafter, supernatants were discarded and pellets were rinsed again with ice-cold diethyl ether (750 µL). Plastic tubes were ventilated to allow for pellets to dry, and these were suspended in CH$_3$CN/H$_2$O [50 µL, 50:50 with 0.1% TFA (v/v)]. Analysis was performed by MALDI (1 µL), UPLC-MS (5 µL) and HPLC (15 µL).

**Microarray Spotting of HDACs 2 and 3 and Assay Linearity Test.** His-tagged HDAC2 and 3 were used at a concentration of 41.4 µM (in 40 mM Tris-HCl pH 8.0, 110 mM NaCl, 2.2 mM KCl, 0.04% (v/v) Tween-20, 20% (v/v) glycerol) and 7.5 µM (in 40 mM HEPES pH 8.0, 110 mM NaCl, 2.2 mM KCl, 20% (v/v) glycerol) respectively. Proteins were diluted to a concentration of 1 µM in a buffer consisting of 2/3 (v/v) DMSO and 1/3 (v/v) saline-sodium citrate (SSC) buffer (150 mM NaCl, 15 mM trisodium citrate, pH 7.0). A 1:2 (v/v) serial dilution of HDAC2 and 3 was prepared in a 384-well plate using a buffer of 2/3 DMSO and 1/3 SSC. Protein solutions were transferred to polystyrene-coated microscope glass slides (Intavis AG, cat. # 54.112) with a Slide Spotter Robot (Intavis AG). Slides were incubated in PBS with 2% (w/v) IGG-free BSA (2.5 mL; 137 mM NaCl, 2.7 mM KCl, 10 mM Na$_2$HPO$_4$, 1.8 mM KH$_2$PO$_4$, pH 7.4) for 60 min at room temperature. Then, slides were incubated at room temperature with Anti-6x-His tag antibody (ThermoFisher Scientific, MA121315, Lot.: UF283577) in PBS with 0.1% BSA (1:5000, 2.5 mL, 15 min), washed with PBS (2.5 mL, 3 × 3 min), incubated with HRP-conjugated goat anti-mouse antibody (ThermoFisher Scientific, 31430, Lot: UJ293428) in PBS with 0.1% BSA (1:5000, 2.5 mL, 30 min), and washed with PBS (2.5 mL, 3 × 3 min). SuperSignal West Femto Maximum Sensitive Substrate was added to the slides (200 µL; ThermoFisher Scientific), and chemiluminescence signal was recorded in a c400 imaging system (Azure, Lowest Sensitivity, 5 min exposure). Binding intensities were evaluated using FIJI including the Microarray Profile addon (OptiNav). The error range and standard deviation were defined by comparing the intensities of two individual spots on the array.

**Recombinant HDAC µSPOT Assays.** µSPOT slides were incubated in HEPES buffer containing 10 mg/mL BSA (2 × 4 mL, 20 min, 25 °C) in 4-well microscope slide tray plates (MoBiTec GmbH, cat. #: 1145-40) and rinsed with standard HEPES buffer (3 × 2 mL). Then, slides were incubated with 6 HDAC concentrations (5-fold dilution; Fig. 3: HDAC1: 77–0.025 nM, HDAC2: 57–0.018 nM, HDAC3/NCoR2: 50–0.016 nM, HDAC8: 56–0.018 nM; Fig. 5a: HDAC1: 50– 0.016 nM, HDAC2: 250–0.016 nM, HDAC3: 50–0.016 nM, HDAC8: 150–0.048 nM) in HEPES buffer (2 mL, 1 h, 25 °C), rinsed (4 × 2 mL), incubated with HRP-conjugated 6x-His tag antibody (1:10000 in HEPES buffer, 2 mL, 1 h, 25 °C), rinsed (4 × 2 mL), and developed. Detection was achieved using SuperSignal West femto maximum sensitivity substrate [ThermoFisher, cat. #: 34095, solution A and B

(300 µL, 1:1)] and a chemiluminescence imaging system (PXi6, Syngene) with 30 s exposure time and no binning. The 6 slides incubated with different HDAC concentrations were imaged together in order to obtain data relative to top and bottom intensity levels. Experiments were performed twice, data were obtained with Array Analyze software (Active Motif), and data analysis and representation were performed with the GraphPad Prism 7 software. Outliers were curated manually and data were fitted to nonlinear regression [log(agonist) vs. response— variable slope (four parameters)]. Curves with top values outside of the 0.8–1.2 range, pEC$_{50}$ values below the studied enzyme concentration range, and incomplete curves were considered weak binders and color-coded with white squares. Non-sigmoidal curves were denoted as not determined (N.D.) The pEC$_{50}$ values were determined as the concentration of HDAC enzyme required for half-maximal binding signal. Data presented in manuscript figures are reported with the number of significant figures allowed by the standard error of the mean (SEM) calculated for each experiment. The raw data in the Supporting Tables are not adjusted corresponding to standard error.

**HDAC Inhibition Assays.** Assays were performed in 96-well plates (Fischer Scientific, cat. #: 3686) in HEPES buffer (final volume: 25 µL/well). Asuha-containing peptide inhibitors (5-fold dilution series, or 50 µM and 5 µM concentration for HDAC4) were incubated with enzyme [HDAC1 (0.5–4 nM), HDAC2 (0.5–3 nM), HDAC3/NCoR2 (1 nM), HDAC8 (0.2–0.5 nM), HDAC4 (0.01 nM) or HDAC6 (0.5–1 nM)] and substrate [HDACs 1–3 and 6: Ac-Leu-Gly-Lys(Ac)-AMC (20 µM), HDACs 4 and 8: Ac-Leu-Gly-Lys(Tfa)-AMC (20 µM and 100 µM, respectively)] for 30 min at 37 °C (AMC: 7-amino-4-methylcoumarin). A solution of trypsin (25 µL, 0.4 mg/mL; final concentration of 0.2 mg/mL) was subsequently added, and the assay was allowed to develop for 15 min at room temperature. Then, fluorescence was recorded and analyzed to afford residual enzymatic activity relative to control wells, assuming a standard fast-on/fast-off mechanism, IC$_{50}$ values were obtained by fitting the resulting data to the dose–response equation with variable Hill slope (Eq. 1). Inhibition $K_i$ values were obtained by using the Cheng-Prusoff equation (Eq. 2) and Michaelis-Menten constants as reported (HDAC1: $K_M$ = 6 µM, HDAC2: $K_M$ = 3 µM, HDAC3: $K_M$ = 6 µM, HDAC6: $K_M$ = 16 µM, HDAC8: $K_M$ = 190 µM)[57]. Assays were performed twice with two internal replicates, and data was analyzed using GraphPad Prism 7 or 8.

$$v_i = v_{bottom} + \frac{v_{top} - v_{bottom}}{1 + 10^{\left(\log IC_{50} - \log[I]\right)h}} \quad (1)$$

$$K_i = \frac{IC_{50}}{1 + \frac{[S]}{K_M}} \quad (2)$$

**HDAC Deacetylation Assays.** Enzyme [HDAC1 (250 nM), HDAC2 (500 nM), HDAC3/NCoR2 (5 nM) or HDAC8 (250 nM)] and Kac peptide (HDACs 1–3: 100 µM, HDAC8: 1 mM) were incubated in HEPES buffer (36 µL) in a 96-well plate for 1 h at 37 °C. Then, a sample from each well (25 µL) was quenched with CH$_3$OH/HCOOH [94:6 (v/v) 12.5 µL] and analyzed by UPLC-MS. Gradients rising linearly 0–10%, 5–15%, 10–20%, or 15–25% during 4 or 9 minutes were employed for separation of starting material and deacetylated product. Mass spectra were used for identification of both species, and conversion was calculated by peak integration at 280 nm. Assays were performed at least twice.

**Cell Treatment and Western Blot.** HEK293T cells (ATCC), provided by the Pless lab at the University of Copenhagen, were cultured under a humidified 5% CO$_2$ atmosphere in Dulbecco's modified Eagle's medium (DMEM, Thermo Scientific, cat. #: 11965118) supplemented with 10% (v/v) fetal bovine serum (FBS), 1% penicillin and 1% streptomycin. Cells were plated into 6-well plates and grown to 80–90% confluency. Treatment with compounds **1a** (10 µM), **13a** (10 µM), SAHA (1 µM), or DMSO was performed in the above-mentioned media for 5 h at 37 °C, after which the media was aspirated, cells were washed with PBS and treated with lysis buffer (1% Triton X-100, 0.2% SDS and cOmplete EDTA-free protease inhibitor cocktail, COEDTAF-RO Sigma-Aldrich, in PBS, 150 µL/well). Upon scraping, lysates were collected in microcentrifuge tubes, sonicated with a Bandelin Sonopuls mini20 (2 s on, 2 s off, 80% amplitude, 1 min), centrifuged (14000 g, 10 min, 4 °C), and protein concentrations of the supernatants were determined by the bicinchoninic acid assay (BCA1, Sigma-Aldrich). The concentrations of the samples were adjusted to the lowest concentration, and 33–54 µg of lysate were mixed with NuPAGE LDS sample buffer (ThermoFisher, NP0007) and sample reducing agent (ThermoFisher, NP0004) followed by heating to 95 °C for 10 min. Samples were then resolved by gel electrophoresis (SDS-PAGE) in NuPAGE gels (12 wells, 4–12% Bis-Tris, 1.0 mm, ThermoFisher, NP0322BOX) with MES running buffer (ThermoFisher, NP000202). Proteins were transferred to PVDF membranes (ThermoFisher, IB24001) using the iBlot2 system and blocked with 5% skim milk in tris-buffered saline with 0.1% tween-20 (TBST) at 25 °C for 1 h. Membranes were then washed with TBST (3 × 5 min), incubated with primary antibody in TBST with 5% BSA (1:1000, 4 °C, overnight), washed with TBST (3 × 5 min), incubated with HRP-conjugated secondary antibody in TBST with 2% skim milk (1:10,000, 25 °C, 1 h), and washed with TBST (2 × 5 min) and TBS (1 × 5 min). Membranes were visualized using enhanced chemiluminescent reagents (Pierce

ECL Western blotting substrate, ThermoFisher, 32106) on a syngene PXi4 image analysis system. Primary antibodies used: acetylated α-tubulin (6-11B-1, sc-23950, Santa Cruz Biotechnology, Inc), vinculin (E1E9V) XP (13901, Lot: 6, Cell Signaling Technology), acetyl-histone H3 (K36) (D9T5Q, 27683, Lot: 1, Cell Signaling Technology), acetyl-histone H3 (K27) (4353, Lot: 1, Cell Signaling Technology). Secondary antibodies used: Anti-rabbit IgG HRP-linked antibody (Cell Signaling Technology, CST-7074S), Anti-mouse IgG HRP-linked antibody (Cell Signaling Technology, CST-7076S). Protein marker used: Precision Plus Protein All Blue Standard (BioRAD, 161-0373).

**Statistics and reproducibility.** All measurements considered were taken from distinct samples. Repeated measurements of a single sample are considered $n = 1$.

**Safety statement.** No unexpected or unusually high safety hazards were encountered.

**Reporting summary.** Further information on research design is available in the Nature Research Reporting Summary linked to this article.

## Data availability
The authors declare that the data supporting the findings of this study are available within the paper and its supplementary information files. Source data are provided with this paper.

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

## Acknowledgements

We thank A.S. Madsen for assistance with data analysis, K. Strømgaard for access to the peptide microarray synthesis and printing equipment, V. Sereikaite for assistance with peptide microarray synthesis, and N. Rajabi for donation of the Fmoc-Asu(*t*Bu)-OH building block. This work was supported by the University of Copenhagen (PhD fellowship to C.M.-Y.), the Carlsberg Foundation (2013-01-0333 and CF18-0442; C.A.O.), the Lundbeck Foundation (Running Cost grant R289-2018-2074), and the European Research Council (ERC-CoG-725172 – SIRFUNCT; C.A.O.).

## Author contributions

Conceptualization, H.M.M. and C.A.O.; Methodology, C.M.-Y., H.M.M., and C.A.O.; Investigation, C.M.-Y., M.B., A.-E.V., and C.S; Writing—Original Draft, C.M.-Y., H.M.M., and C.A.O.; Writing - Review & Editing, C.M.-Y., M.B., A.-E.V., C.S., H.M.M., and C.A.O.; Visualization, C.M.-Y.; Supervision, H.M.M. and C.A.O.; Project Administration, C.A.O.; Funding Acquisition, C.A.O.

## Competing interests

The authors declare no competing interests.
