## [Peer Review File · Nature Communications]

REVIEWER COMMENTS

Reviewer #1 (Remarks to the Author):

In their manuscript, Moreno-Yruela et al. utilize hydroxamic acid-modified histone microarrays to interrogate sequence specificity of class I HDAC isoforms. Overall, this is a very interesting tool that could be in principle used for the design of isoform-specific HDAC inhibitors, chemoproteomic probes or to define specificities of physiological HDAC complexes in the future.

While the presented data are mostly solid, I believe that more thorough data analysis and more elaborate corroboration by in-solution experiments would better convince readers of the general applicability of the described experimental platform and significantly strengthen the impact of presented findings.

Major points:

1. The microarrays are said to provide (semi)quantitative data (binding affinities). For this to be true, it would be required that within the experimental window, the observed chemiluminescence signal linearly correlates with the amount of bound HDAC. The authors shall provide such experimental data by spotting dilution series of HDACs (preferred) or His-tag mimicking peptides and define the linear range of the assay. It shall be also noted that using fluorescently labeled anti-His mAbs (or better fluorescently labeled HDACs) could improve the assay linearity compared to the chemiluminescence detection that is prone to oversaturating signals.
2. Throughout the arrays, binding of HDAC3 is substantially more prevalent. It can in principle reflect the wide substrate specificity of HDAC3 but can be also an artefact of the assay that would make selection of “specific sequences” somewhat difficult (see also 5). Based on available published data (substrate specificities of class I HDACs) the authors might want to discuss in detail this observation. Alternatively, an experiment with spotted HDACs (as above) might provide more experimental insights.
3. Within the text it is stressed that the arrays can be used to map sequence specificities of class I HDACs. However, no general sequence preferences for individual HDACs are shown, and no thorough analysis is presented. The authors shall analyze their data in detail and provide sequence logo plots to highlight sequence preferences at individual peptide positions. Again, these findings shall be compared to published data.

4. Figs 4A, S1: “Titration of single amino acid frame-shifted peptides and single amino acid truncation variants defined the exact binding sites and the relative binding affinities. The mapping of these sites revealed a complex sequence requirement for HDACs 1, 2, and 8 – at least for residues within ± 7 positions relative to the acetylated lysine – and highlighted the importance of the relative position of the acetylated lysine within the peptide.” The observed differences in peptide affinity can be in principle attributed to a single amino acid deletions/frame shifts as suggested. Alternatively, however, these can also be a consequence of steric interference of the microarray matrix as it looks like there is a noticeable trend towards lower affinities of peptides with the Asuha function placed closer to the C-terminus of the peptide. To verify claims related to these Figures (and disprove the latter scenario), it would be critical to corroborate microarray data in in-solution experiments (IC50 measurements) using the same set of peptides as in Fig 4A.

5. Page 11 – “Notably, our microarray approach afforded a number of peptide candidates with unique inhibition profiles across HDAC subtypes, suggesting that similar approaches may facilitate the discovery of inhibitors and peptide-based tools with tailored HDAC selectivity.” Indeed, in the microarray format, many peptides really reveal pronounced selectivity for a single individual HDAC isoform, so in principle it shall be possible to design highly selective isoform inhibitors. Yet, no single-isoform selective inhibitor has been reported by the authors. The microarray derived peptide selectivity, i.e. a specific inhibition of a single (or maximum two) isoform shall be shown for a selected set of peptides in in-solution experiments. Several such candidate peptides (e.g., H2B-K85X for HDAC3, H4-K77X for HDAC2/3) can be easily picked up from heat maps in Fig. 3.

Minor points:

6. Page 3: “Conventional display of acetylated peptides in microarray format does not allow to study HDAC action due to the fast turnover and resulting transient nature of the HDAC–peptide interaction“. As HDACs are deacetylases, I would argue the opposite, i.e., that conventional microarrays are much better suited for studying HDAC action (as opposed to binding).

7. Fig. 2B – by comparing Asuha and Asuapa zinc-binding groups, it looks like the latter may be more responsive for changes in peptide sequences. Why were the Asuha peptides selected for microarrays?

8. Discussion: “Peptide microarrays are broadly applied to study HATs and histone readers, especially bromodomains. Their application to HDAC “erasers”, however, has remained more limited and required secondary, time-consuming readouts”. The presented assay comprises three steps: HDAC treatment-primary antibody binding-(chemiluminescence) detection. Virtually identical setup:

HDAC treatment-primary antibody binding-(fluorescence) detection is typically used for microarray activity-based assays so both approaches are likely comparable in terms of experimental efforts.

9.Fig. 4C, pg 8 – “Taken together, our results highlight the applicability of hydroxamic acid-modified microarrays to interrogate HDAC substrate recognition with unprecedented throughput.” While for the 2c peptide, an increase in binding affinity is observed for all HDACs (compared to 1c) in the microarray format, the opposite trend is observed for deacetylation of corresponding 2a/1a peptides. Consequently, I would advise an extreme caution in inferring the substrate specificity from the microarray binding signals.

Reviewer #2 (Remarks to the Author):

This is a fascinating study with great implications for further use. My major concern is the use of the concentration of the enzymes, there is no information on the purity or specific activity. The authors should use a promiscuous substrate to correlat concentrations and specific activities of the different substrate.

A competition experiment with a selective inhibitor for a given subtype and a non-inhibitor would further validate the robustness. Furthermore, competition experments with small molecules would further validate use for inhibitor screens. A further advancement would be to show tolerability of cell permeation sequences. The editor should advise on the relevance of these suggestions for acceptance.

Point-by-point explanation of changes made in response to the “Reviewer’s Comments to Author”

Reviewer #1 (Remarks to the Author):

In their manuscript, Moreno-Yruela et al. utilize hydroxamic acid-modified histone microarrays to interrogate sequence specificity of class I HDAC isoforms. Overall, this is a very interesting tool that could be in principle used for the design of isoform-specific HDAC inhibitors, chemoproteomic probes or to define specificities of physiological HDAC complexes in the future.

While the presented data are mostly solid, I believe that more thorough data analysis and more elaborate corroboration by in-solution experiments would better convince readers of the general applicability of the described experimental platform and significantly strengthen the impact of presented findings.

Major points:

1. The microarrays are said to provide (semi)quantitative data (binding affinities). For this to be true, it would be required that within the experimental window, the observed chemiluminescence signal linearly correlates with the amount of bound HDAC. The authors shall provide such experimental data by spotting dilution series of HDACs (preferred) or His-tag mimicking peptides and define the linear range of the assay. It shall be also noted that using fluorescently labeled anti-His mAbs (or better fluorescently labeled HDACs) could improve the assay linearity compared to the chemiluminescence detection that is prone to oversaturating signals.

This is a great suggestion. We agree with the reviewer in that, even though chemiluminescence from HRP-linked antibody signals are generally considered linearly dependent on the local concentration of antigen, it is good practice to evaluate assay linearity when a new setup is introduced. We addressed this by spotting (or ‘dot-blotting’) a dilution series of His tag HDAC2 protein on our microarray surface. Gratifyingly, the anti-His₆ antibody detection used in our histone microarrays provided a clear linear correlation for HDAC2 signal versus spotted amount, as depicted in Supplementary Figs. 1a and 1b. The following sentences have been added to the manuscript in order to address this matter (page 5):

The detection of a dilution series of His-tagged HDAC2 spotted on a μ SPOT slide indicated a linear correlation between signal and spotted amount (Supplementary Fig. 1, $r^2 = 0.96$) when a horseradish peroxidase (HRP)-conjugated anti-His antibody was used. Thus, taken together, the μ SPOT method allowed for economic production of hundreds of microarray copies from a single round of synthesis and thereby facilitated the semi-quantitative assessment of HDAC isozyme-peptide interactions (Fig. 1c)³⁷⁻³⁸.

2. Throughout the arrays, binding of HDAC3 is substantially more prevalent. It can in principle reflect the wide substrate specificity of HDAC3 but can be also an artefact of the assay that would make selection of “specific sequences” somewhat difficult (see also 5). Based on available published data (substrate specificities of class I HDACs) the authors might want to discuss in detail this observation. Alternatively, an experiment with spotted HDACs (as above) might provide more experimental insights.

Indeed, HDAC3 binding was overall more prevalent than that of other HDACs, which we also find interesting. It appears that HDAC3 can recognize Asuha in most contexts, being less sensitive to the underlying peptide sequence. This relates to the fact that HDAC3 shows similar efficiencies towards substrates with differences in the peptide sequence and even small variations in the acyl modification (Gurard-Levin et al., 2010; Madsen et al., 2012; McClure et al., 2017). In addition, HDAC3 does not show the interference with the microarray matrix observed for HDACs 1, 2 and 8 (see answer to 4, and Supplementary Fig. 7), which could explain the observed higher rate of positive results. Prompted, by this comment we have synthesized additional peptides and tested their potency in order to extend the validation and assess HDAC3 data further (see answer to 5). The following sentences have been added to the manuscript (page 6):

HDACs 1, 2 and 8 displayed high sequence specificity, in agreement with previous substrate selectivity studies on HDAC8^{28,45}. In contrast, HDAC3 bound more promiscuously and to sequences located both at the core and the tails of the histones. This might indicate that HDAC3 is less sensitive to the peptide sequence, as also reported for short peptide substrates⁴⁶⁻⁴⁷ and acetylated peptide microarrays²⁰.

And page 8:

This could indicate that particularly HDACs 1, 2 and 8 are sensitive to the steric effect of the microarray matrix and struggle to bind to Asuha residues closer than 5 amino acids to the C-terminal peptide conjugation site. On the other hand, HDAC3 was not affected by this effect, which could also explain the higher rate of positive results for this isozyme.

3. Within the text it is stressed that the arrays can be used to map sequence specificities of class I HDACs. However, no general sequence preferences for individual HDACs are shown, and no thorough analysis is presented. The authors shall analyze their data in detail and provide sequence logo plots to highlight sequence preferences at individual peptide positions. Again, these findings shall be compared to published data.

We thank the reviewer for this excellent suggestion. We have now included Supplementary Fig. 2 showing sequence logo plots for each of the class I HDACs (see also reply to comment 2). This analysis reveals a preference for Arg, Lys and Gly residues in most of the positions before and after Asuha, and with just minor differences between specific HDACs. This is in agreement with the preferred HDAC substrates for in vitro assays (e.g. LGK(Ac), TARK(Ac), RHKK(Ac), ...), and

also with previous studies on HDAC8 (Gurard-Levin, 2008) and HDAC3 (Gurard-Levin, 2010). However, we agree that this analysis would be more elaborate if performed with a larger library of randomized peptides which we considered out of the scope of this manuscript. In the current array design, we focused entirely on histone sequences, which could result in the bias towards Arg, Lys and Gly amino acids next to AsuHa and hide potential isozyme-specific preferences in these positions. The following sentences have been added to the manuscript to include this additional analysis (page 6):

Sequence logo plots were generated to highlight the amino acid preferences of each HDAC (Supplementary Fig. 2). These plots revealed overall preference for Arg and Lys residues at the positions flanking the AsuHa residue for all class I HDACs, which was previously observed for HDAC8²⁸ and, to some extent, also HDAC3²⁰. Interestingly, HDAC6 shows opposite preference²². Further, class I HDAC selectivity was observed for sequences containing Gly at the -2 and -3 positions, and Arg and Lys at the -4, +3 and +4 positions. Not surprisingly, histone tails are rich in Arg, Lys and Gly residues, which could explain the obtained sequence preference. In this regard, we expect that future evaluation of potential isozyme-specific preferences would benefit from the use of randomized peptide arrays.

4. Figs 4A, S1 (now Supplementary Fig. 3): “Titration of single amino acid frame-shifted peptides and single amino acid truncation variants defined the exact binding sites and the relative binding affinities. The mapping of these sites revealed a complex sequence requirement for HDACs 1, 2, and 8 – at least for residues within ± 7 positions relative to the acetylated lysine – and highlighted the importance of the relative position of the acetylated lysine within the peptide.” The observed differences in peptide affinity can be in principle attributed to a single amino acid deletions/frame shifts as suggested. Alternatively, however, these can also be a consequence of steric interference of the microarray matrix as it looks like there is a noticeable trend towards lower affinities of peptides with the AsuHa function placed closer to the C-terminus of the peptide. To verify claims related to these Figures (and disprove the latter scenario), it would be critical to corroborate microarray data in in-solution experiments (IC50 measurements) using the same set of peptides as in Fig 4A.

We thank the reviewer for pointing this out, as steric interferences indeed seem to have an effect. We have now resynthesized five of the peptides included in Fig. 4a, including the parent peptide (**1a**), a frame-shifted peptide and a truncated peptide without the C-terminal part (**2a** and **3a**), and a frame-shifted peptide and a truncated peptide without the N-terminal part (**4a** and **5a**). We then obtained inhibitor constants against HDACs 1-3 and 8, and observed that peptides **4a** and **5a** were 10-fold less potent than the parent peptide (which could only be predicted for HDACs 2 and 8) and that peptides **2a** and **3a** retained potency for all HDACs (Fig. 4c and Supplementary Fig. 7). This indicates that the loss of binding affinity observed for HDACs 1, 2 and 8 on all H4K12AsuHa peptides without C-terminal fragments could result from steric effects on the microarray, and that these enzymes cannot bind to AsuHa when it is close to the C-terminus of the sequence. The same effect could be observed later for peptides **13a** and **17a** (Fig. 6). Thus lending an explanation

for most of the observed outliers/discrepancies and further suggesting that longer linkers may further improve the predictive value of our microarray approach. The results have been modified as follows (page 8):

Titration of single amino acid frame-shifted peptides and single amino acid truncation variants defined the exact binding sites and indicated that the C-terminus of the selected peptide is required for binding to HDACs 1, 2 and 8 (Supplementary Figs. 3-6). Peptides **1a-5a** were then synthesized, purified and tested as HDAC inhibitors to validate these observations. Peptides **4a** and **5a** showed 10-fold lower potency than the parent peptide (**1a**), as suggested by the data on HDACs 2 and 8, but not HDAC1 (Fig. 4c and Supplementary Fig. 7). More strikingly, peptides **2a** and **3a** with C-terminal truncations retained potency against all HDACs. This could indicate that particularly HDACs 1, 2 and 8 are sensitive to the steric effect of the microarray matrix and struggle to bind to Asuha residues closer than 5 amino acids to the C-terminal peptide conjugation site. Overall, the mapping of the H4K12 site revealed a complex sequence requirement for HDACs 1, 2 and 8, especially for residues within ± 7 positions relative to the acetylated lysine, and it highlighted steric effects introduced by the immobilization and thus the importance to validate results in solution.

(page 12):

Peptides **13a** and **17a**, for example, appeared to portray preference for HDACs 2 and 3 over HDAC1 on the array, but exhibited similar potencies against all three isoforms in solution. This highlighted once more the importance of sufficient C-terminal spacing in microarray format, as both peptides included the Asuha residue within 5 amino acid positions to the linker.

And the following sentence has been added to the discussion (page 15):

In particular, the potency of peptides with the Asuha residue close to the C-terminus were not well predicted, likely due to steric interference of the microarray matrix.

5. Page 11 – “Notably, our microarray approach afforded a number of peptide candidates with unique inhibition profiles across HDAC subtypes, suggesting that similar approaches may facilitate the discovery of inhibitors and peptide-based tools with tailored HDAC selectivity.” Indeed, in the microarray format, many peptides really reveal pronounced selectivity for a single individual HDAC isoform, so in principle it shall be possible to design highly selective isoform inhibitors. Yet, no single-isoform selective inhibitor has been reported by the authors. The microarray derived peptide selectivity, i.e. a specific inhibition of a single (or maximum two) isoform shall be shown for a selected set of peptides in in-solution experiments. Several such candidate peptides (e.g., H2B-K85X for HDAC3, H4-K77X for HDAC2/3) can be easily picked up from heat maps in Fig. 3.

We agree with the reviewer in that more extensive validation of inhibitors in solution would help to evaluate the scope of the technique and at the same time strengthen the concept that peptide scaffolds can achieve improved or even novel selectivity profiles. In order to address this and

previous points, we resynthesized four additional peptides from the screening in Fig. 3 (peptides **11a**, **12a**, **13a** and **17a**) and corresponding controls (peptides **11b**, **12b**, **13b** and **17b**). The dose-response inhibition curves are presented in Fig. 6a and Supplementary Fig. 9, peptides **11a** and **13a** are shown in Fig. 6b, and inhibitor constants have been added to Fig. 6c.

Peptides **11a** and **12a**, based on H2B-K85X, and **13a**, based on H3-K18X, were selected as potential HDAC3-selective inhibitors as suggested by the reviewer. As predicted, all of them showed preference for HDAC3 vs. HDACs 1 and 2. Peptide **11a** was 4-fold selective vs. HDACs 1 and 2 and 63-fold selective vs. HDAC8. Even larger preference was observed for peptide **13a** (8-fold vs. HDAC1, 28-fold vs. HDAC2), although this peptide also inhibited HDAC8 potently (data point not determined in the microarray). On the other hand, peptide **17a** based on H4-K77X was tested as potential HDAC2/3-selective inhibitor, but it proved to inhibit HDACs 1, 2 and 3 to a similar extent. This is interesting, as peptide **17a** has the Asuha residue three positions away from the C-terminus and could have suffered from steric hindrance on the microarray (see also answer to 4). Fig. 6 has been extended, and the results section has been modified to include the new peptides. In particular, the following sentences have been added (page 10):

The determined inhibitor profiles confirm the potency of multiple hits and validate the concept that peptide scaffolds can achieve isozyme selectivity. Sequences **11a**, **12a** and **13a** exhibited selectivity for HDAC3 in the microarray (Fig. 3). For the resynthesized peptides, we found ~4-fold, ~2-fold and >8-fold selectivity for HDAC3 vs. HDACs 1 and 2, respectively (Figs. 6c and 6d), which supports the use of our screening technique to discover selective inhibitor leads. Peptide **11a** was also >60-fold selective for HDAC3 vs. HDAC8, whereas peptide **13a** inhibited HDAC8 potently, which was not anticipated.

And page 12 (see also answer to 4):

Peptides **13a** and **17a**, which appeared to exhibit preference for HDACs 2 and 3 over HDAC1 in the array experiments, were both equipotent across all three recombinant HDACs 1–3, when tested in solution. This highlighted once more the importance of sufficient C-terminal spacing in microarray format, as both peptides included the Asuha residue within 5 amino acid positions of the linker

Minor points:

6. Page 3: “Conventional display of acetylated peptides in microarray format does not allow to study HDAC action due to the fast turnover and resulting transient nature of the HDAC-peptide interaction“. As HDACs are deacetylases, I would argue the opposite, i.e., that conventional microarrays are much better suited for studying HDAC action (as opposed to binding).

This is an excellent observation. The corresponding sentence has been modified as follows (page 3):

Conventional display of acetylated peptides in microarray format does not allow studying HDAC binding due to the fast turnover and resulting transient HDAC-peptide interaction.

7. Fig. 2B – by comparing Asuha and Asuapa zinc-binding groups, it looks like the latter may be more responsive for changes in peptide sequences. Why were the Asuha peptides selected for microarrays?

In our initial warhead screening, we focused on obtaining the most binding signal, which we obtained with the Asuha building block. This allowed for microarray interrogation at a low concentration and for dose-response experiments with a reasonable consumption of enzyme. At that stage, we observed sequence dependency on the signals obtained from both Asuha- and Asuapa-containing peptides, as shown later in the complete Asuha microarray. Nevertheless, it would be interesting to repeat the study using Asuapa as the selected building block and compare the performance and sequence discrimination of both microarrays. The following sentences have been added to the results (page 4):

Both Asuha and Asuapa provided sequence-dependent binding results, and Asuha was selected for further investigations due to the higher HDAC retention shown by the corresponding peptides.

And to the discussion (page 15):

This question could be addressed using the approach presented herein, together with alternative modifications such as the here described Asuapa building block.

8. Discussion: “Peptide microarrays are broadly applied to study HATs and histone readers, especially bromodomains. Their application to HDAC “erasers”, however, has remained more limited and required secondary, time-consuming readouts”. The presented assay comprises three steps: HDAC treatment-primary antibody binding-(chemiluminescence) detection. Virtually identical setup: HDAC treatment-primary antibody binding-(fluorescence) detection is typically used for microarray activity-based assays so both approaches are likely comparable in terms of experimental efforts.

We agree with the reviewer in that our assay is as time-consuming as microarray activity-based assays, as both require incubation with HDAC, primary antibody binding, and detection. However, we consider our setup advantageous for measuring binding of the HDAC to the peptide, since acetylated peptides would need to be cross-linked to the enzyme or require a fluorophore-dependent binding readout. In addition, when antibodies are used to read possible peptide modifications, the quality of the readout will be limited by the ability of the antibody to recognize such modifications without any bias for the corresponding sequence context. This is a major limitation across all peptide microarray applications for post-translational modification readout. The corresponding sentence has been modified in order to clarify this (page 14):

Their application to erasers of histone acetylation, however, has remained limited to activity-based approaches, as microarray binding experiments required secondary, time-consuming readouts¹⁹⁻²⁹. In addition, activity-based approaches rely on antibodies specific for a certain peptide modification, which can sometimes lead to biases induced by the sequence context. Here, we describe a facile, cost-effective, and scalable protocol for the preparation of the necessary hydroxamic acid building block and demonstrate its use to produce hydroxamic acid-modified peptide microarrays. These arrays allow the simple and direct profiling of HDAC binding and inhibition with minimal sample consumption, and with the use of tag antibodies extensively validated.

9. Fig. 4C, pg 8 – “Taken together, our results highlight the applicability of hydroxamic acid-modified microarrays to interrogate HDAC substrate recognition with unprecedented throughput.” While for the 2c peptide, an increase in binding affinity is observed for all HDACs (compared to 1c) in the microarray format, the opposite trend is observed for deacetylation of corresponding 2a/1a peptides. Consequently, I would advise an extreme caution in inferring the substrate specificity from the microarray binding signals.

This is an excellent point that we have also come to realize. Indeed, we believe only strong effects on substrate recognition can be inferred from the microarray data (e.g. peptides **1a/c** compared to peptides **3a/c**). We have modified the following sentences to address this in the results (page 10):

Taken together, our results highlight the applicability of hydroxamic acid-modified microarrays to interrogate major changes in HDAC substrate recognition with unprecedented throughput.

And in the discussion (page 15):

We found that sequence requirements for the recognition of histone tails by HDACs 1, 2, and 8 remain conserved to a certain extent for the Asuha-modified peptides, thus allowing for prediction of major changes in the turnover of acetylated substrates in response to PTMs at other sites.

Reviewer #2 (Remarks to the Author):

This is a fascinating study with great implications for further use. My major concern is the use of the concentration of the enzymes, there is no information on the purity or specific activity. The authors should use a promiscuous substrate to correlat concentrations and specific activities of the different substrate.

We thank the reviewer for pointing this out, as the information was indeed missing in the experimental section. The following details have been added to the Materials and Methods (page 16):

HDAC1 (full length, C-terminal His tag, C-terminal FLAG tag, BPS Bioscience, cat. #: 50051, lots 170105-1 and 181108-1, purity \geq 79%, activity \geq 460 pmol/min/ μ g), HDAC2 (full length, C-terminal His tag, BPS Bioscience, cat. #: 50002, lots 160701 and 160630, purity \geq 84%, activity \geq 675 pmol/min/ μ g), HDAC3/NCoR2 (full length, C-terminal His tag, NCoR2 N-terminal GST tag, BPS Bioscience, cat. #: 50003, lots 130819 and 190327, purity \geq 80%, activity \geq 3000 pmol/min/ μ g), HDAC8 (full length, C-terminal His tag, BPS Bioscience, cat. #: 50008, lots 150714 and 161216, purity \geq 90%, activity \geq 300 pmol/min/ μ g), HDAC4 (aa 627-1084, BPS Bioscience, cat. #: 50004, lot 130828-G, purity \geq 89%, activity \geq 103225 pmol/min/ μ g), HDAC6 (full length, BPS Bioscience, cat. #: 50056, lot 151130-C, purity \geq 88%, activity \geq 150 pmol/min/ μ g).

A competition experiment with a selective inhibitor for a given subtype and a non-inhibitor would further validate the robustness. Furthermore, competition experiments with small molecules would further validate use for inhibitor screens.

We agree with the reviewer in which microarray data could be further improved by adding inhibitors as competitors of HDAC binding. In our case, we decided to evaluate subtype selectivity with in-solution inhibitor assays for a selected number of resynthesized hits, but the proposed experiment would be useful for future screening. The following sentence has been modified in the discussion (page 15):

Moreover, incubation with recombinant HDACs bearing active site mutations or co-incubation with validated HDAC probes could improve the selection of competitive inhibitor candidates from the array screening.

A further advancement would be to show tolerability of cell permeation sequences.

We thank the reviewer for the suggestion, as it prompted us to test cell permeation for selected microarray peptides. In particular, **1a** and **13a** showed potency on living cells at 10 μ M concentration, measured as upregulation of α -tubulin acetylation (HDAC6 inhibition) and of histone acetylation (HDACs 1-3 inhibition). Both peptides harbor exclusively basic or hydrophobic residues, which may have helped to achieve sufficient cellular uptake even without potentially toxic or otherwise interfering cell permeation sequences. Fig. 7 has been added showing sample Western blots and quantification of protein acetylation, and full blots and raw data have been added as Supporting Fig. 12 and Source data, respectively. In addition, the following section has been added to the results (page 14):

Hydroxamic acid-modified peptides exhibit activity in cells

Next, we aimed to explore the cellular activity of selected HDAC inhibitors. We chose peptides **1a** and **13a**, which exhibit nanomolar potency against class I HDACs and HDAC6 and, at the same

time, harbor multiple basic residues that could facilitate cell permeation. HEK293T cells were treated with the peptides for 5 h, and subsequent cell lysis and Western blot analysis revealed upregulation of acetylation of both tubulin and histones compared to DMSO control (Fig. 7a). These results strongly indicate that peptides **1a** and **13a** effectively inhibit the cytosolic HDAC6 and the nuclear HDACs 1-3 in the cell, which is in agreement with their *in vitro* inhibition profiles against recombinant HDACs. In particular, peptide **13a** induced an 8-fold increase in tubulin acetylation and a 4-fold increase in H3K36 acetylation (Fig. 7b). Thus, the high-throughput screening and in-solution evaluation delivered cellularly active HDAC probes.

And the discussion has been extended with the following sentence (page 16):

Selected microarray hits proved to penetrate the cell membrane and elicit hyperacetylation of class I and class IIb HDAC targets.

The editor should advise on the relevance of these suggestions for acceptance.

REVIEWERS' COMMENTS

Reviewer #1 (Remarks to the Author):

In the revised manuscript, the authors provided a substantial amount of the experimental data to address most points raised. While additional data (and corresponding changes to the manuscript text) definitely strengthened some aspects of the manuscript, these, unfortunately, at the same time did not allay my concerns about the predictive power of the microarray towards the design of isoform-specific inhibitors/substrates. These problems are exemplified by Figs. 4, 6, and S7.

Using Fig. 6d as an example, it looks like no correlation exists between microarray and in-solution inhibitor potency. In fact, the most potent inhibitors for HDAC1 (11a), HDAC2 (11a), and HDAC8 (13a), as determined by in-solution experiments, did not show any appreciable binding in the microarray format. As such, no predictions can be made based on the microarray data. Similarly, over 40-fold difference is observed for IC50s of 16a and 17a against HDAC8, yet the microarray data suggest their equipotent binding.

It is up to the handling editor whether to accept the manuscript even with the limited (no) correlation between microarray and in-solution experiments. The absence of correlation is, in my opinion, the major problem and microarrays are not well suited to identify/predict new HDAC inhibitors/substrates as claimed. However, if the manuscript is deemed suitable for NatComm, I would suggest that the authors rephrase affected passages to provide more balanced interpretation of the data, as these might be somewhat overinterpreted in the current form (e.g., abstract, page 11).

Additional points:

1. Fig. 3 – knowing the importance of the C-terminal spacing for HDAC1/2 binding, it would be helpful to label potentially affected peptides in the figure (e.g., asterisk?).
2. Fig. 5 – as some of PTMs are close the C-terminus of peptides (less accessible to HDACs1/2), their importance/predictive power might also be limited due to spatial requirements.
3. Fig. 6d – it would be helpful to visualize correlation (or the absence thereof) between microarray and in-solution data in a graphical format.

Reviewer #2 (Remarks to the Author):

The authors provide an extensive revision which addresses concerns and remarks, can be accepted now.

Point-by-point explanation of changes made:

Reviewer #1 (Remarks to the Author):

In the revised manuscript, the authors provided a substantial amount of the experimental data to address most points raised. While additional data (and corresponding changes to the manuscript text) definitely strengthened some aspects of the manuscript, these, unfortunately, at the same time did not allay my concerns about the predictive power of the microarray towards the design of isoform-specific inhibitors/substrates. These problems are exemplified by Figs. 4, 6, and S7.

Using Fig. 6d as an example, it looks like no correlation exists between microarray and in-solution inhibitor potency. In fact, the most potent inhibitors for HDAC1 (11a), HDAC2 (11a), and HDAC8 (13a), as determined by in-solution experiments, did not show any appreciable binding in the microarray format. As such, no predictions can be made based on the microarray data. Similarly, over 40-fold difference is observed for IC50s of 16a and 17a against HDAC8, yet the microarray data suggest their equipotent binding.

It is up to the handling editor whether to accept the manuscript even with the limited (no) correlation between microarray and in-solution experiments. The absence of correlation is, in my opinion, the major problem and microarrays are not well suited to identify/predict new HDAC inhibitors/substrates as claimed. However, if the manuscript is deemed suitable for NatComm, I would suggest that the authors rephrase affected passages to provide more balanced interpretation of the data, as these might be somewhat overinterpreted in the current form (e.g., abstract, page 11).

We understand the concerns of the reviewer regarding the predictive power of the microarray presented. In order to address them, we have rephrased the abstract:

Here, we synthesize hydroxamic acid-modified histone peptides and use them in femtomolar microarrays for the direct capture and detection of the four class I HDAC isozymes. Follow-up functional assays in solution demonstrate their suitability to discover HDAC substrates and inhibitors with nanomolar potency and activity in cellular assays.

the introduction (page 3):

Here, we report the synthesis and application of peptide microarrays that circumvent the aforementioned limitations by introduction of hydroxamic acid-containing residues in place of the Kac residues (Fig. 1b)³²⁻³⁴, which thereby provide a platform to investigate HDAC binding preferences in high-throughput (Fig. 1c). Complementary functional assays evaluate the ability of this approach to predict HDAC binding, inhibition, and activity.

[modified from: "Here, we report the synthesis and application of peptide microarrays that circumvent the aforementioned limitations by introduction of hydroxamic acid-containing residues in place of the Kac residues (Fig. 1b)³²⁻³⁴, which thereby provide a platform to dissect investigate HDAC function binding preferences in high-throughput (Fig. 1c). Complementary functional assays demonstrate evaluate the value ability of this approach to predict HDAC binding, inhibition, and activity."]

the results section (page 12):

The determined inhibitor profiles confirm the potency of multiple hits and validate the concept that peptide scaffolds can achieve isozyme selectivity, but they also reveal that potent inhibitors can be overlooked during microarray screening.

In addition, peptide **11a**, which was among the most potent inhibitors of HDACs 1 and 2, did not show tight binding on the array for these two isozymes.

and the discussion (page 15):

Here, we systematically explored the sequence and PTM requirements for the recognition of histone tails, as represented by Asuha-modified peptides, by HDACs 1, 2, 3 and 8.

[modified from: "Here, we systematically explored the predictive power of this modification. We found that sequence and PTM requirements for the recognition of histone tails, as represented by Asuha-modified peptides, by HDACs 1, 2, 3 and 8 remain largely conserved for the Asuha-modified peptides, thus allowing for prediction of major changes in the turnover of acetylated substrates in response to PTMs at other sites."]

and

The potency of peptides with the Asuha residue close to the C-terminus were not well predicted, likely due to steric interference of the microarray matrix, and a number of additional potent inhibitors were overlooked by microarray screening.

Additional points:

1. Fig. 3 – knowing the importance of the C-terminal spacing for HDAC1/2 binding, it would be helpful to label potentially affected peptides in the figure (e.g., asterisk?).

We thank the reviewer for this suggestion that will be very helpful for the reader. The corresponding peptides have been labeled in Figure 3 with the following footnote:

[†]Matrix steric effects might impede protein binding to these peptides.

2. Fig. 5 – as some of PTMs are close the C-terminus of peptides (less accessible to HDACs1/2), their importance/predictive power might also be limited due to spatial requirements.

We thank the reviewer for pointing this out. Since HDACs 1, 2 and 8 presented low microarray affinity for peptides with Asuha close to the C-terminus, it could be inferred that PTMs at those positions might not be interrogated correctly on the array. In order to highlight this, the following sentence has been added to the manuscript (page 11):

Modifications at Lys23 also diminished HDAC8 binding, although its proximity to the C-terminus makes it difficult to interpret this trend due to the aforementioned matrix steric effects.

In addition, Lys14 on the (1–16) peptide and Lys23 on the (10–25) peptide are marked with a footnote in Figure 5a to highlight potential steric effects:

Matrix steric effects might contribute to the effects at these positions.

3. Fig. 6d – it would be helpful to visualize correlation (or the absence thereof) between microarray and in-solution data in a graphical format.

This is a very good suggestion. We have now prepared scatter plots to visually compare the microarray affinity and inhibitor potency data presented in Fig. 6d. This has been included as Supporting Fig. 12 and is referenced both in the main text:

Though a number of the potencies recorded for resynthesized peptides recapitulated the microarray findings, the potency of a number of the weaker binders was not estimated accurately in the microarray screening, thereby highlighting the need for detailed follow-up validation when applying this technology (Supplementary Fig. 12).

and in the legend of Fig. 6:

See Supplementary Fig. 12 for visual representation.

Reviewer #2 (Remarks to the Author):

The authors provide an extensive revision which addresses concerns and remarks, can be accepted now.